ary

# Why we publish where we do: Faculty publishing values and their relationship to review, promotion and tenure expectations

Meredith T. Niles[1]*, Lesley A. Schimanski[2], Erin C. McKiernan[3], Juan Pablo Alperin[2]*

1 Department of Nutrition and Food Sciences & Food Systems Program, University of Vermont, Burlington, VT, United States of America, 2 Scholarly Communications Lab, Simon Fraser University, Vancouver, Canada, 3 Departamento de Física, Universidad Nacional Autónoma de México, Mexico City, Mexico

* mtniles@uvm.edu (MTN); juan@alperin.ca (JPA)

**Data Availability Statement:** RPT documents from this project can be found here: Alperin, Juan Pablo; Muñoz Nieves, Carol; Schimanski, Lesley; McKiernan, Erin C.; Niles, Meredith T., 2018,

## Abstract

Using an online survey of academics at 55 randomly selected institutions across the US and Canada, we explore priorities for publishing decisions and their perceived importance within review, promotion, and tenure (RPT). We find that respondents most value journal readership, while they believe their peers most value prestige and related metrics such as impact factor when submitting their work for publication. Respondents indicated that total number of publications, number of publications per year, and journal name recognition were the most valued factors in RPT. Older and tenured respondents (most likely to serve on RPT committees) were less likely to value journal prestige and metrics for publishing, while untenured respondents were more likely to value these factors. These results suggest disconnects between what academics value versus what they think their peers value, and between the importance of journal prestige and metrics for tenured versus untenured faculty in publishing and RPT perceptions.

## 1. Introduction

The concept of "publish or perish" has been a dominant credo in academia, especially in high-income Western contexts, for decades, but its effects may be particularly evident as the rate of academic publishing continues to grow rapidly. Between 2006 and 2016, the number of academic publications increased 56% [1]. In 2018, there were more than 33,000 academic peer-reviewed English language journals publishing more than three million articles a year [2]. This ever increasing volume of research has led many academics to question how to keep up with this pace of knowledge communication [3].

While these trends pose obvious challenges to those trying to stay abreast of the latest developments in their field, they may also be having more subtle consequences for academia writ large, as they touch not just on the practice of research, but on the very nature of academic careers. Namely, the increased volume of academic publishing may influence how academics

"Terms and Concepts found in Tenure and Promotion Guidelines from the US and Canada", https://doi.org/10.7910/DVN/VY4TJE, Harvard Dataverse, V3, UNF:6:PQC7QoilolhDrokzDPxxyQ== [fileUNF]. Niles, Meredith T.; Schimanski, Lesley A.; McKiernan, Erin C.; Alperin, Juan Pablo, 2020, "Data for: Why we publish where we do", https://doi.org/10.7910/DVN/MRLHNO, Harvard Dataverse, V1.

**Funding:** Funding for this project was provided to JPA, MTN, ECM, and LAS from the Open Society Foundations (OR2017-39637) The funders had no role in study design, data collection and analysis, decision to publish, or preparation of the manuscript.

**Competing interests:** MTN is a member of the board of directors of The Public Library of Science (PLOS). This role has in no way influenced the outcome or development of this work or the peer-review process, nor does it alter our adherence to PLOS ONE policies on sharing data and materials.

perceive academic publishing expectations. Faculty at academic institutions assume that strong research and publication records are necessary in their review, promotion, and tenure (RPT) process [4, 5, 6]. Furthermore, faculty express concerns about the amount and type of publishing expected of them (i.e., that it should be in prestigious journals with high journal impact factors (JIF)) and their capacity to achieve the amount of publications expected by their universities [7]. Indeed, some universities undertake interventions to increase faculty publishing efforts, in part because of the potential financial gains associated with this increased volume [8]. Prioritization of quantity and journal metrics have also led many to question and study the quality of research outputs [9], as retractions of articles, especially in "high impact" or prestigious journals increases [10], and reproducibility of results are in question [11].

Amidst this increasing volume of literature and the potential consequences that come with it, this study aimed to explore the drivers of academic faculty publishing decisions, particularly as they relate to the RPT process. Using a dataset gathered from faculty of 55 institutions across the US and Canada, we asked:

1. Do faculty perceive measures of impact, prestige, and volume to influence their decisions on where to publish their academic work?

2. In what ways do faculty perceive their own publishing decision-making as different from that of their peers?

3. How do faculty perceive the valuation of their publication outputs and metrics in the RPT process?

4. What is the relationship between faculty publishing decisions and their perceptions of the RPT process?

## 2. Methods

### 2.1 Survey and data collection

To answer these questions, we surveyed faculty from a broad set of universities in the United States and Canada, as part of a larger project on current RPT practices [12, 13]. For this project, we collected RPT documents (e.g., policies, guidelines, presentations) from a representative sample of universities in the United States and Canada, and many of their academic units (e.g., faculty, department, school). The sample of institutions was stratified based on institution type using the 2015 edition of the Carnegie Classification of Institutions of Higher Education [14] and the 2016 edition of the Maclean's University Rankings [15], which classify institutions into those focused on doctoral (i.e., research-intensive) programs (R-Type), those that predominantly focus on master's degrees (M-Type), and those focused on undergraduate (i.e., baccalaureate) programs (B-Type). Full details of the sample selection and document collection strategy are available in Alperin et al. [12].

Following this strategy, we were able to obtain documents from 381 academic units of 60 universities (out of a set of 129 universities for which we obtained university-level documents). Using this list of academic units, we searched for a page listing the faculty members of each unit, and selected up to five faculty members from without paying attention to their characteristics. In the end, we were able to identify 1,644 faculty from 334 of the 381 units spanning all 60 institutions (with some units not listing email addresses publicly, and some units not having 5 faculty members listed). We chose to limit the collection of names from the units for which we had guidelines so that, in a forthcoming study, we could study the relationship between the two.

The selected participants were invited to participate in an online survey on September 17th, 2018, with reminders sent on a weekly basis until October 29th, 2018 to any who had not yet responded. A total of 338 people (22%) from 55 different institutions responded to the survey. Of these, 84 (25%) were faculty at Canadian institutions and the remaining 254 (75%) were from the United States; 223 (66%) were from R-Type institutions, 111 (32%) from M-Type institutions, and 4 (1%) from B-Type institutions. Responses were then anonymized, leaving only the institution type and discipline along with the survey responses for analysis, as per the research protocol filed with the Office of Research Ethics at Simon Fraser University (file number: 2018s0264).

## 2.2 Data analysis and model development

Data were aggregated into Stata 15.0 [16] for analysis. To analyze statistically significant differences between variables, we selected appropriate statistical tests based on the distribution of data including the Kruskal Wallis test, chi square tests, Wilcoxon Rank sum test, and Spearman's correlations for non-parametric data and one-way analysis of variance and Pearson's correlations for continuous data.

To understand how multiple factors relate to publication decisions, we constructed ordered logistic regression models across the ten publication factors with multiple key independent variables including demographic factors (age, gender, institution type, tenure status), total number of publications the respondent typically published annually (pubs published), and a sub-set of components perceived to be valued by the respondent in the RPT process that were related to publishing (e.g., rptpubnumbers, rptpre-print, rptopenaccess, rptsociety, rptjournalIF, rptjournalname, rptpubtotal) (Table 1). Models are reported in log-odds statistics, which can be interpreted as coefficients greater than 1 indicating a greater odds of occurrence and coefficients less than 1 indicating a reduced odds of occurrence.

# 3. Results

## 3.1 Survey respondents overview

The largest portion of survey respondents were tenured faculty (63.5%), followed by tenure-track who were not yet tenured, (20.3%), Department chairs (8.8%), Deans (3.8%), research faculty (1.9%), and Lecturers (1.6%). Given that Department chairs and Deans are typically positions held by individuals later in their careers, these two responses were added to the tenured faculty category for further analyses (bringing the total to 76.1%). Similarly, research faculty and lecturers were grouped with not-yet-tenured faculty. The overwhelming majority of respondents reported a PhD as their highest degree (92.9%), while 5.1% reported a professional degree and 2.1% reported a master's degree. Our sample was nearly perfectly split between men (49.9%) and women (49.3%) with a small portion of respondents (less than 1%) indicating non-binary identity. Due to the very small number of those reporting non-binary gender identity, the data from these participants were necessarily excluded from those statistical analyses/models that differentiated between gender categories.

Just over two thirds of respondents (67.6%) came from R-type institutions, while the remaining 32.4% came from M-type institutions. Four responses were received from B-type institutions, and given this small sample size, they are not considered in our statistical analysis. We classified the respondents' academic units by discipline using the National Academies Taxonomy [17] and found that 53% came from Social Sciences and Humanities (SSH), 21% from Life Sciences (LS), 17% from Physical Sciences and Mathematics (PSM); and the remaining 9% from units that could not be classified into a single area.

**Table 1. Variable questions and scales used in analysis.**

| Variable Type | Variable | Question | Scale |
|---|---|---|---|
| Demographic | age | *How old are you?* | 1 = Under 18, 2 = 18–24, 3 = 25–34, 4 = 35–44, 5 = 45–54, 6 = 55–64, 7 = 65+ |
| | gender | *Which best describes your gender identity?* | 1 = male, 0 = female |
| | r-type | *Categorized by institution type (not asked of respondents)* | 1 = R-type, 0 = M-type |
| | tenure status | *Which of the following best describes you?* | 1 = Tenure-track faculty (tenured), Department Chair, Dean; 0 = Tenure-track faculty (pre-tenure), Research faculty (non-tenure track), Lecturer or primarily teaching position |
| Publication Rate | pubs published | *Which of the following best describes your academic peer-reviewed publication history (e.g., journal articles, monographs, book chapters, conference proceedings?)* | 1 = No peer-reviewed publications per year; 2 = Less than 1 peer-reviewed publication per year; 3 = 1–2 peer-reviewed publications per years; 4 = 3–5 peer-reviewed publications per years; 5 = More than 6 peer-reviewed publications per year |
| Publication Importance Factors | *How important are the following factors to you/to your colleagues for deciding where you/your colleagues submit your academic work for publication?* | | 1 = Not important, 6 = Very important |
| | merit pay | Receive direct support (e.g., merit pay or additional funding) for publications in specific journals | |
| | readership | Has a readership that I want to reach | |
| | journal IF | Impact factor of the journal | |
| | society journal | Journal of a society to which I belong | |
| | journal read | Journal/publisher/venue that I regularly read | |
| | journal peers | Journal/publisher/venue that my peers regularly read | |
| | journal cited | How often the journal appears to be cited | |
| | journal prestige | Overall prestige of the journal/publisher/venue | |
| | open access | That the publication makes (or allows me to make) my article freely available to the public | |
| | journal cost | The cost (or lack of cost) to publish | |
| RPT perceptions | *To what extent do you believe the following are valued for your performance reviews?* | | 1 = Not valued, 6 = Very valued |
| | rpt blog | Blog posts or other publication communication outputs | |
| | rpt book chapter | Book chapters | |
| | rpt book | Book publications or monographs | |
| | rpt pub numbers | Number of publications per year | |
| | rpt performance | Performances or artistic outputs | |
| | rpt media | Popular media coverage of my work | |
| | rpt preprint | Pre-prints | |
| | rpt open access | Public availability of the journals (i.e., open access) | |
| | rpt society | Society journal publications | |
| | rpt journal IF | The impact factor of the journals | |
| | rpt journal name | The name recognition of the journals | |
| | rpt pub total | Total number of publications | |

The majority of respondents (65.3%) had served on a RPT committee previously, with tenured faculty much more likely to have served on RPT committees (84% compared to 13% non-tenured, $p < 0.001$). Older faculty were also more likely to have served on RPT committees ($p < 0.001$).

## 3.2 Factors affecting publication decisions

Respondents predominantly averaged 1–2 peer-reviewed publications per year (47.4%), followed by 3–5 publications (23.2%), less than one-peer-reviewed publication per year (18.0%), more than six peer-reviewed publications per year (8.7%), and 2.8% of respondents not publishing peer-reviewed publications. Women reported publishing fewer articles than men (p = 0.084) (S2 Table). Respondents at R-type institutions were also more likely to publish than those at M-type institutions (p < 0.001) (S2 Table).

There were clear factors considered important by respondents when evaluating where to publish their academic work (Fig 1). Overall, respondents' top three most valued factors were: (1) whether the journal had a readership they wanted to reach, (2) the overall prestige of the journal/publisher/venue, and (3) whether it was a journal/publisher/venue that their peers regularly read. Some demographics correlated with variability on these values (S1 and S2 Tables). Non-tenured respondents placed higher importance on the JIF compared to tenured faculty (mean 4.61 compared to 4.18, p = 0.029). The rated importance of the JIF (r = -0.156, p = 0.009), journal citation frequency (r = -0.182, p = 0.002), and journal prestige (r = -0.165, p = 0.005) were negatively correlated with age (i.e., were less important to older respondents) while that of society journals was positively correlated with age (r = 0.124, p = 0.039). Finally, journal cost was a more important factor for women than for men (mean 4.14 compared to 3.16, p = 0.001).

Compared to their own perceptions of important priorities when publishing, respondents perceived differences in how their peers rate important factors for publishing (Fig 2, Table 2). Considering the mean responses, the top factors respondents thought their peers felt were important included: (1) the overall prestige of the journal/publisher/venue, (2) the JIF, and (3) both the readership they want to reach and the journal/publisher/venue being regularly read by their peers. Overall, we find that there are many statistically significant differences between how people perceive their own publishing priorities versus those of their peers. For example, respondents were more likely to think their peers valued the prestige of the journal/publisher/venue compared to themselves (mean 5.02 others compared to 4.76 self, p = 0.013), as well as to value the JIF compared to themselves (mean 4.77 others compared to 4.29 self, p < 0.001), and how often the journal is cited (mean 4.57 others, 3.87 self, p < 0.001). Conversely, respondents were more likely to perceive they valued the readership compared to their peers (mean 5.02 self compared to 4.60 others, p < 0.001), and that the publication was open access (mean 3.29 self compared to 2.73 others, p< 0.001).

## 3.3 Perceptions of performance, review, and tenure

Respondents perceived certain factors were valued in the RPT process more than others. Overall, regardless of demographics, respondents perceived that the total number of publications (mean 5.40), the number of publications per year (mean 5.29), and the name recognition of the journals (mean 4.83) were the most valued factors in their RPT processes (Fig 3).

Perceived values of particular factors in the RPT process varied according to a number of demographics (S4, S5 and S6 Tables). Correlations between age and such factors suggest that older faculty value blogs (r = 0.160, p = 0.010), book chapters (r = 0.122, p = 0.045), performances (r = 0.317, p < 0.001), and open access journals (r = 0.250, p < 0.001) in the RPT process more than younger faculty. Comparing tenured and non-tenured respondents, books (mean 4.34 compared to 3.74 non-tenured, p = 0.025) and book chapters (mean 3.61 tenured compared to 3.14 non-tenured, p = 0.009) were more valued by those who were tenured. Finally, women valued publications per year more than did men (mean 5.51 for women compared to 5.11 for men, p = 0.001), and also total number of publications (mean 5.60 for women

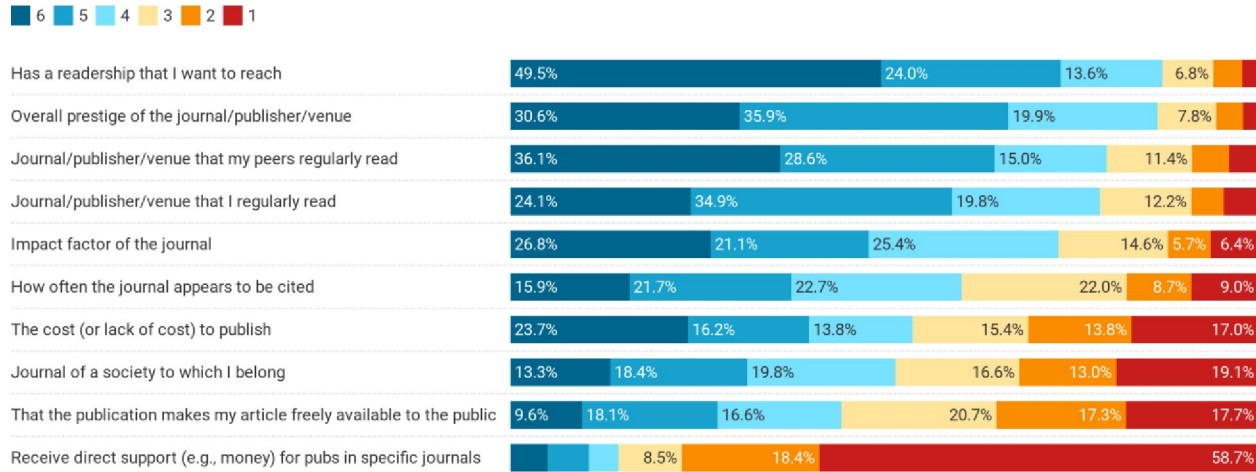

**Fig 1. Importance of various factors when respondents consider where to submit their academic work for publication.** Scale ranges from 1 (not important) to 6 (very important). Factors are ordered in their overall rate of importance (percent indicating a 4, 5 or 6).

compared to 5.20 for men, p = 0.001). We also found that respondents at R-type institutions, when compared with those at M-type institutions, were more likely to place higher importance on journal name recognition (mean = 4.97 compared to 4.58, p = 0.013) and JIF (mean = 4.81 compared to 4.37, p = 0.014) and less likely to place importance on book chapters (mean = 3.29 compared to 3.86, p = 0.001).

## 3.4 Publication decision models

To examine the factors related to publication decisions, we ran a series of ordered logit models with the ten publication priorities (as listed in Fig 1) as dependent variables (outcomes) and demographics, publication history, and perceptions of factors that matter in the RPT process as independent variables. The question here was, for each factor that influences publication decisions, what was the relative importance of demographics, publication history, and perception of the RPT process in determining the importance placed on that factor? For instance, if

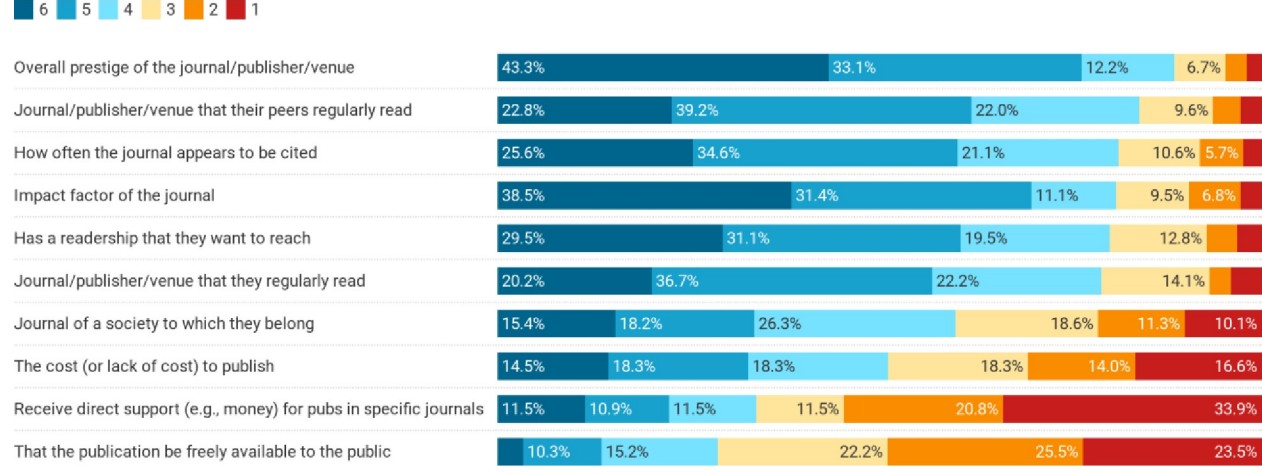

**Fig 2. Importance of various factors respondents think their peers consider when submitting their academic work for publication.** Scale ranges from 1 (not important) to 6 (very important). Factors are ordered in their overall rate of importance (percent indicating a 4, 5 or 6).

**Table 2. Respondents' mean ratings of factors affecting publication decisions compared to the mean rating of their perceptions of how their peers would rate the same factors.** Factors are ordered from greatest to least difference between self and peer perceptions. Higher means for a given variable are highlighted for emphasis.

| Variable | Self Mean | Peer's Mean | p value |
|---|---|---|---|
| Receive direct support (e.g., money) for pubs in specific journals | 1.94 | 2.79 | <0.001 |
| How often the journal appears to be cited | 3.87 | 4.57 | <0.001 |
| That the publication makes my article freely available to the public | 3.29 | 2.73 | <0.001 |
| Impact factor of the journal (JIF) | 4.29 | 4.77 | <0.001 |
| Has a readership that I/they want to reach | 5.02 | 4.60 | <0.001 |
| Journal of a society to which I belong | 3.45 | 3.77 | 0.023 |
| Overall prestige of the journal/publisher/venue | 4.76 | 5.02 | 0.013 |
| The cost (or lack of cost) to publish | 3.70 | 3.51 | 0.241 |
| Journal/publisher/venue that my peers regularly read | 4.68 | 4.60 | 0.488 |
| Journal/publisher/venue that I regularly read | 4.48 | 4.45 | 0.790 |

respondents value the JIF when selecting where to disseminate their work, is this best explained by, for example, their age, their gender, their number of prior publications, or their perceived value of the JIF within the context of RPT evaluations? Such models allow us to consider all of these factors simultaneously in seeking to understand the values guiding faculty publishing decisions. All model results are reported in the supplementary materials (S5–S14 Tables), and here we explore the general trends found across the models through a summary table (Fig 4).

Across the ten models we find that the factors affecting publication decisions are more likely to correlate with the perception of what is valued in the RPT process than with demographic factors including age, gender, institution type, and tenure status. In fact, demographic factors only have a statistically significant relationship in two of the 10 models. In one of these (model 4), older people have increased odds (b = 1.37, p = 0.019) of valuing society journals in publication decisions and in another (model 10) men have reduced odds (b = 0.40, p = 0.001) of finding cost important for publication decisions (i.e., women are more likely to find cost important in publication decisions). In another model (model 2), it is not demographic characteristics, but faculty behavior (the number of peer-reviewed publications per year) that results in increased odds of faculty valuing journal readership (b = 1.68, p = 0.003).

Conversely, we find that in 9 of the 10 models (all but model 10) at least one aspect of faculty's perception of the RPT process is correlated with a factor affecting publication decisions. Perceptions of the importance of journal name (i.e., name recognition of the journals) and open access value (e.g., public availability of the journals) in the RPT process are the factors most frequently associated with a publication decision model (3 of the 10 models), followed by perceived importance of pre-prints and JIF in RPT (2 of the 10 models). Overall, these results suggest that faculty perceptions of the RPT process are a greater influence on publication decisions than are university type or other respondent demographics.

## 4. Discussion

Through our survey of faculty at more than 50 institutions across the US and Canada, we explored factors related to publishing decisions and their relationship to the RPT process. We found that overall, respondents value journal readership, journal/publisher prestige and whether the journal will be read by their peers. At the same time, respondents felt that their peers prioritized factors differently when considering where to publish, namely that their peers put greater emphasis on the journal's prestige, JIF and journal citations. We found that tenure

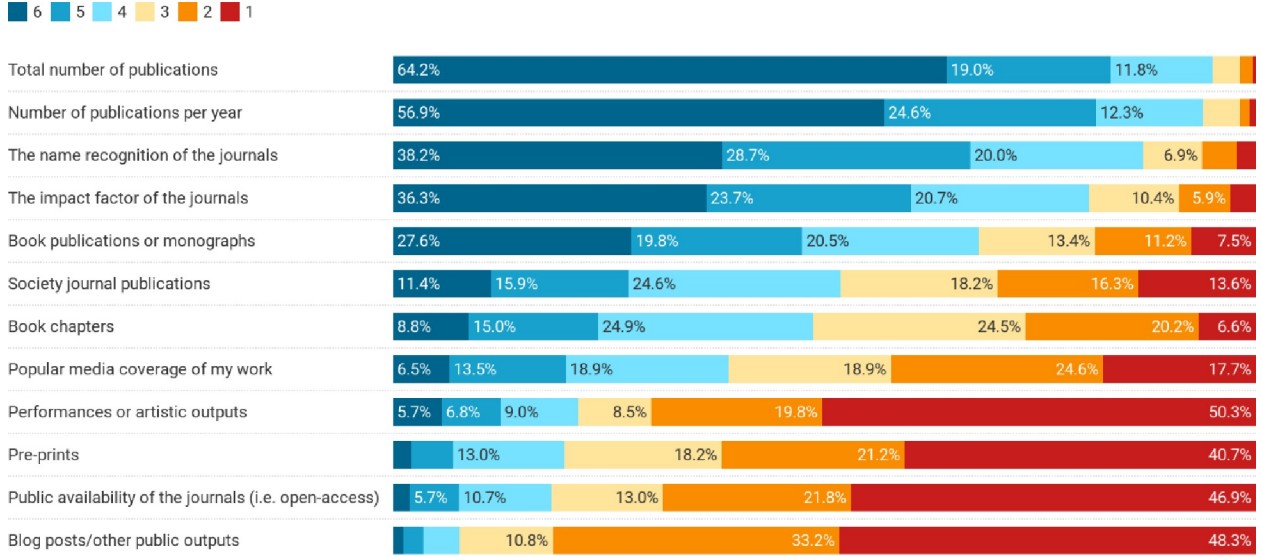

**Fig 3. Perceived value of factors in the RPT process.** Bars show percentage of respondents. Scale ranged from 1 (not valued) to 6 (very valued). Factors are ordered in their overall rate of importance (e.g., percent of respondents indicating a 4, 5 or 6).

status and age are important distinguishers in these perspectives, as older and tenured faculty place less emphasis on the JIF, how often the journal appears to be cited, and the overall prestige of the journal/publisher/venue than do their younger and non-tenured colleagues.

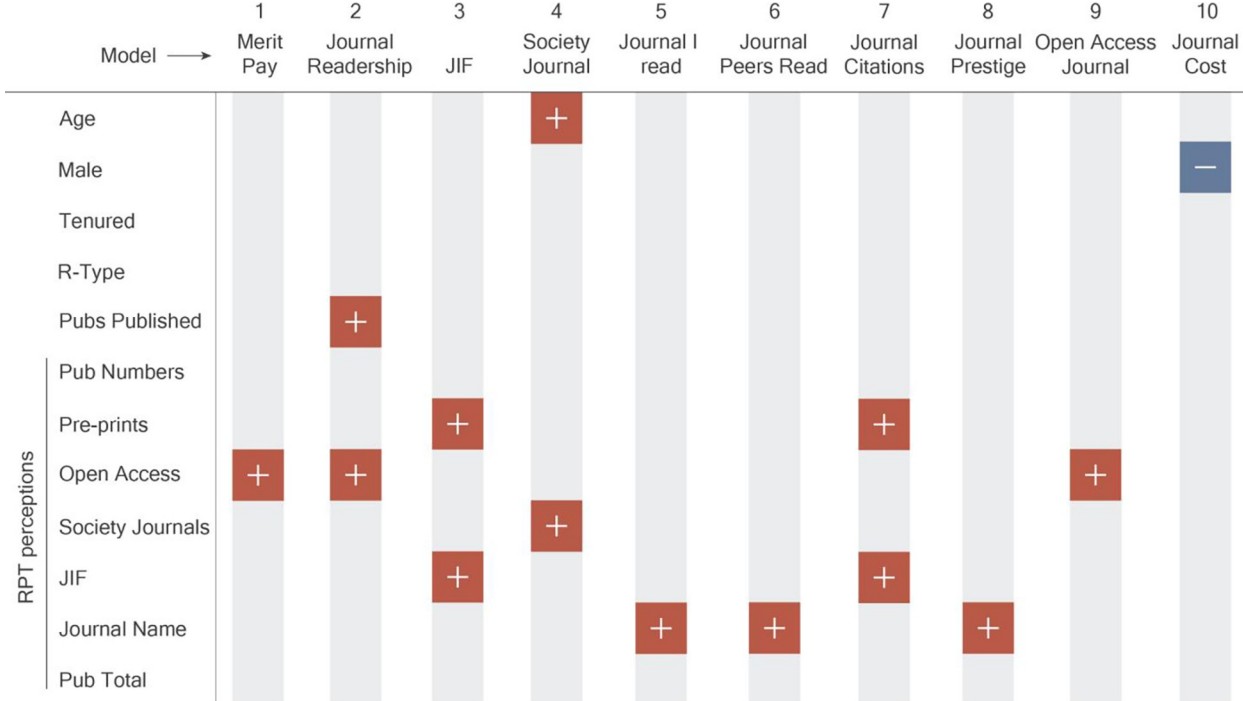

**Fig 4. Publication decision model outputs.** Dependent variables are in the first column, with independent variables across the top row. Positive symbols indicate a significant greater odds relationship. For example, in model 7 (Journal Citations) below, there is a greater odds relationship with the JIF and pre-prints, which means that respondents who felt JIF and pre-prints are important in the RPT process had greater odds of valuing journal citations in publication decisions. Conversely, negative symbols indicate a reduced odds relationship with the dependent variable. Full model results can be found in S7–S16 Tables.

When it comes to faculty perceptions of the RPT process, respondents overwhelmingly expressed that they thought publication quantity and prestige were the most important, with total number of publications, number of publications per year, and the name recognition of the journal perceived as most valuable. However, this perception was not held equally between respondents of all ages and career stages, nor by respondents at different institution types. Older and tenured faculty were less likely to place emphasis on these factors than younger and non-tenured faculty, and respondents from M-type institutions were less likely to place importance on the journal's name recognition or its JIF. When looking at how these perceptions, demographics, and the institutional characteristics affect publication decisions through a series of ordered logit models, we find that it is the RPT perceptions that are more frequently linked to publication decisions versus any institutional or demographic factors.

These results confirm that the RPT process, and faculty's perceptions of it, have an important role in shaping where faculty publish. These perceptions may in part be driven by the RPT documents themselves; in a recent study, we found that 40% of R-type institutions mentioned the JIF in their RPT documents, and 87% of the institutions mentioning it did so in a way that encouraged JIF consideration in the RPT decision [13]. Similarly, we found journal name recognition and the JIF to be among the most important factors shaping publication decisions, especially by respondents at R-type institutions. Whether RPT documents play a role in shaping these perceptions or not, our findings show that how the RPT process is perceived matters in shaping faculty decisions about where to publish. That being said, understanding the values that drive publication decisions is complicated by the mismatch between faculty's own values and how they perceive those of their peers, whom they see as valuing prestige and the JIF more than they do.

Our results confirm previous findings that faculty seem to be often driven by readership and peer exposure to their work when deciding where to publish [18], but simultaneously add depth to discussions about the role that prestigious journal names and citation measures like the JIF have in shaping publication decisions. Most importantly, our work suggests that any shift away from JIF, journal names or citation measures may be challenged not by faculty's own values, but by the perception they have of their peer's publication decisions, which we find to be markedly different than their own. Put plainly, our work suggests that faculty are guided by a perception that their peers are more driven by journal prestige, journal metrics (i.e., JIF and journal citations), and money (i.e., merit pay) than they are, while they themselves value readership and open access of a journal more.

The idea that respondents generally perceive themselves in a more favorable light than their peers (e.g., less driven by prestige or money), elicits multiple self-bias concepts prevalent in social psychology, including illusory superiority [19]. That people generally perceive themselves to be "better" is not unique to this particular topic [20, 21]. However, our results do suggest that the guise of fame and prestige in academic publishing may not matter as much as previously thought. As such, subjective norms—how we perceive what others value or think, and the perceived social pressure to act in a certain way [22]—could be critical to enabling understanding of people's individual preferences compared to their peers. If faculty truly value journal metrics and prestige outcomes less than readership and peers reading their work, but perceive "others" to be the promoters of these concepts, fostering conversations and other activities that allow faculty to make their values known may be critical to addressing the disconnect. Doing so may enable faculty to make publication decisions that are consistent with their own values.

The perception of others' values is especially important when we consider that the disconnect is also apparent in what faculty perceive is valued in the RPT process. We find that faculty, especially non-tenured faculty, perceive that quantity and prestige are major drivers of the

RPT process. Others have found similar outcomes with faculty reporting that publication in high ranking journals [23, 24], or quantity of publications [23] rather than quality of publications [25] are among the most important factors for determining academic career progression. However, our results suggest that these perceptions may be counter to reality, since we also find that older and tenured faculty—those most likely to serve on RPT committees—value these factors less and are significantly more likely to value outputs such as blogs and open access journals, results that are consistent with other findings [18]. Thus, non-tenured faculty may be driven by traditional scholarly incentives, which they believe to be valued in the RPT process, leading to behavioral patterns that are inconsistent with their expressed drivers of publication decisions. More than 60% of early career faculty strongly agreed that they shape their publication decisions to match those perceived as important for RPT [18], a finding consistent with our multiple models that showed RPT perceptions are a significant factor in various publishing decisions (Fig 4). For example, while we find that faculty generally perceive themselves to value open access publications more than their peers, they also perceive this is not highly valued in the RPT process. This potential mismatch between an individual's values and the perceptions of others' values, including those doing evaluations in RPT committees, may explain the incongruence between the enthusiasm for open access publishing and faculty's actual behaviors [18].

Furthermore, respondents ranked components of publishing that have to do with publicly available outputs (e.g., pre-prints, open access, and blogs) as the least important in the RPT process. These perceptions may be in part due to the lack of attention that such public facing documents receive and the extent to which they are promoted (or not) in RPT documents. Alperin et al. [12] found that RPT documents generally lacked focus on public facing outputs such as these. For example, mentions of open access only appeared in the RPT documents of five percent of the 129 institutions they sampled, and the majority of those were cautious or neutral, and not supportive of open access publishing venues. In this context, our results indicate that the lack of emphasis on such outputs may be related to perceptions that they are of less importance in the RPT process.

Lastly, we would be remiss not to highlight that these mismatches in perception have a potentially outsized impact on women in academia. We found that women were significantly more likely than men to publish fewer articles and consider the cost of a journal in their publication decisions, and more likely than men to value the number of publications per year and total number of publications with regard to the RPT process. There are multiple potential factors at play with these results. First, others have also found that women publish less than their male counterparts (e.g. [26, 27]), which have been correlated with research resources that have been historically lower for women in institutions [26]. In science, technology, engineering, and math (STEM) fields, men are also invited to submit publications twice as frequently as women [28]. Second, these perceptions may be driven in part by how men and women spend their time in academia, as previous research has found that women are more likely to work additional hours devoted to teaching and may produce fewer research papers [29, 30]. Relatedly, men are more likely to rate research as important to their career advancement than are women, suggesting that women may be turning away from research in their careers [31] even though they perceive that research is valued in the RPT process. However, it is also possible that women are more cost sensitive and value the number of publications because in some cases, women submit fewer grants [32, 33] and also receive fewer grants [34]. Since grants provide funding for publications, and often include number of publications as a criteria of evaluation, this may explain some of our findings.

We also want to highlight that there are several limitations to the scope and interpretation of this work. First, we acknowledge that the geographic focus area in North America, and

especially Canada and the U.S. means that this work may not be representative of other regions, especially non-English speaking or Western regions. As well, given that the survey utilizes self-reported information, we acknowledge that these are perceptions, which may not reflect actual behaviors. Future research could better connect individual responses to such questions with actual publication records to better verify the links between self-reported behaviors and actual publication decisions.

## 5. Conclusion

As the pace of academic publishing continues to grow, so do the concerns about a focus on quantity over quality by individual faculty and by universities as a whole, through the RPT process. Our results confirm that faculty value the readership of a journal over other citation metrics or perceived prestige, but that such values may be at odds with what they believe to be valuable in the RPT process. However, our work goes further by showing that these same faculty believe that quantity and prestige of publications still dominate RPT decisions and that faculty, especially those who are non-tenured and younger, believe these factors to be the most important, even though the very people serving on RPT committees value these outputs far less. The resulting mismatches are concerning, especially when coupled with the increased volume of research, as it suggests that the factors guiding publication decisions are inconsistent with faculty's own values.

Our earlier analysis similarly found that values related to various concepts of 'publicness' were significantly present in RPT documents, signaling an institutionalized valuation of publicly oriented activities beyond academic publishing, but that faculty may not feel they will be rewarded if they pursue them, as the documents simultaneously presented clear guidelines to publish traditional research outputs and to use citation metrics to assess them [12, 13]. The results presented here confirm that faculty perceive these publicly oriented outputs (e.g., blogs, pre-prints, and open access) as being far less important in the RPT process than other traditional research metrics and outputs. All this to say, it appears there is a continued need to hold conversations in academia about the nature of academic publishing and how publishing decisions are perceived in the RPT process.

These conversations should consider that, in an environment in which there is a growing number of ways in which faculty can share their work, and in which there is an ever increasing number of works available, many faculty are most interested in choosing academic publishing venues that have a readership of interest and find journal metrics or other factors related to prestige and monetary incentives less important. Importantly, they should also consider that faculty perceive their peers to place more value on journal metrics, prestige, and monetary incentives than themselves, but that, despite these personal motivations, the majority of them believe it is publication quantity and journal prestige and metrics that are the most heavily weighted factors in the RPT process.

These findings can also be brought to bear on the conversations that are already taking place regarding the need for alternative models for research evaluation. Efforts such as HuMetricsHSS (Humane Metrics in the Humanities and Social Sciences) and DORA (Declaration on Research Assessment) have recognized a need to change the values underlying the evaluation of academic outputs. Our results indicate that the value of such visible efforts and public discussions about how to evaluate research may be in helping faculty realize that their peers share their values, rather than in changing the values themselves. We suggest that future research could explicitly evaluate how such alternative approaches are utilized by faculty, and whether they are serving to change what faculty perceive will be valued in the RPT process.

## Supporting information

**S1 Table. Mean responses, standard error (SE) and p values for publishing decisions and productivity by tenure status.** ANOVA were used for statistical significance tests. (DOCX)

**S2 Table. Mean responses, standard error (SE) and p values for publishing decisions and productivity by gender and institution type.** ANOVA were used for statistical significance tests for all variables except for "pubs published", which is a categorical variable where chi2 tests were used. (DOCX)

**S3 Table. Spearman's correlations and p values for publishing decisions and productivity by age.** (DOCX)

**S4 Table. Spearman's correlations and p values for perception of the RPT process by age.** (DOCX)

**S5 Table. Mean responses and p values for perception of the RPT process by tenure status.** ANOVA were used for statistical significance tests. (DOCX)

**S6 Table. Mean responses and p values for perception of the RPT process by gender and institution type.** ANOVA were used for statistical significance tests. (DOCX)

**S7 Table. Ordered logistic model predicting merit pay as a factor in publishing decisions (Model 1).** Total n = 154. (DOCX)

**S8 Table. Ordered logistic model predicting readership respondents want to reach as a factor in publishing decisions (Model 2).** Total n = 203. (DOCX)

**S9 Table. Ordered logistic model predicting journal impact factor as a factor in publishing decisions (Model 3).** Total n = 203. (DOCX)

**S10 Table. Ordered logistic model predicting society journals as a factor in publishing decisions (Model 4).** Total n = 204. (DOCX)

**S11 Table. Ordered logistic model predicting journal/venue/publisher they regularly read as a factor in publishing decisions (Model 5).** Total n = 205. (DOCX)

**S12 Table. Ordered logistic model predicting journal/venue/publisher their peers regularly read as a factor in publishing decisions (Model 6).** Total n = 203. (DOCX)

**S13 Table. Ordered logistic model predicting how often a journal is cited as a factor in publishing decisions (Model 7).** Total n = 203. (DOCX)

**S14 Table. Ordered logistic model predicting overall prestige of the journal/venue/publisher as a factor in publishing decisions (Model 8).** Total n = 205.
(DOCX)

**S15 Table. Ordered logistic model predicting public availability of the publication (i.e. open access) as a factor in publishing decisions (Model 9).** Total n = 202.
(DOCX)

**S16 Table. Ordered logistic model predicting journal cost to publish as a factor in publishing decisions (Model 10).** Total n = 192.
(DOCX)

## Acknowledgments

We would like to thank Jennifer Peruniak and Carol Muñoz Nieves for their assistance in survey design and data collection, and Esteban Morales and Michelle La for their contributions to our team. We would also like to thank and acknowledge the OpenCon community who brought us together in the first place, and whose work inspires and invigorates us year after year.

## Author Contributions

**Conceptualization:** Meredith T. Niles, Lesley A. Schimanski, Erin C. McKiernan, Juan Pablo Alperin.

**Data curation:** Meredith T. Niles, Juan Pablo Alperin.

**Formal analysis:** Meredith T. Niles.

**Funding acquisition:** Meredith T. Niles, Lesley A. Schimanski, Erin C. McKiernan, Juan Pablo Alperin.

**Methodology:** Meredith T. Niles, Lesley A. Schimanski, Erin C. McKiernan, Juan Pablo Alperin.

**Project administration:** Juan Pablo Alperin.

**Visualization:** Meredith T. Niles.

**Writing – original draft:** Meredith T. Niles, Erin C. McKiernan, Juan Pablo Alperin.

**Writing – review & editing:** Meredith T. Niles, Lesley A. Schimanski, Juan Pablo Alperin.

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
