## [Decision Letter · Decision Letter 0]

7 Oct 2019

PONE-D-19-21675

Why we publish where we do: Faculty publishing values and their relationship to review, promotion and tenure expectations

PLOS ONE

Dear Alperin,

Thank you for submitting your manuscript to PLOS ONE. After careful consideration, we feel that it has merit but does not fully meet PLOS ONE’s publication criteria as it currently stands. Therefore, we invite you to submit a revised version of the manuscript that addresses the points raised during the review process.

One expert in the field has commented on your paper and, although their valuation of the paper is positive, it is true that several issues contained in the current version of the paper (and queriers raised from its review) need revision and response from you. You will find many comments -all of them should be responded in detail- that are appended below.

We would appreciate receiving your revised manuscript by Nov 15 2019 11:59PM. To enhance the reproducibility of your results, we recommend that if applicable you deposit your laboratory protocols in protocols.io, where a protocol can be assigned its own identifier (DOI) such that it can be cited independently in the future. For instructions see: http://journals.plos.org/plosone/s/submission-guidelines#loc-laboratory-protocols

We look forward to receiving your revised manuscript.

Kind regards,

Sergio A. Useche, Ph.D.

Academic Editor

PLOS ONE

Journal Requirements:

1. Please do not include funding sources in the Acknowledgments or anywhere else in the manuscript file. Funding information should only be entered in the financial disclosure section of the submission system. https://journals.plos.org/plosone/s/submission-guidelines#loc-acknowledgments

2. If materials, methods, and protocols are well established, authors may cite articles where those protocols are described in detail, but the submission should include sufficient information to be understood independent of these references (https://journals.plos.org/plosone/s/submission-guidelines#loc-materials-and-methods). In this case, please provide additional information about your sampling method (from https://elifesciences.org/articles/42254) in this submission's methods section in order to enable reproducibility and replicability.

3. Thank you for including your competing interests statement; "MTN is a member of the board

of directors of The Public Library of Science (PLOS). This role has in no way influenced the outcome or development of this work or the peer-review process."

Reviewers' comments:

Reviewer's Responses to Questions

**Comments to the Author**

1. Is the manuscript technically sound, and do the data support the conclusions?

Reviewer #1: No

2. Has the statistical analysis been performed appropriately and rigorously? 

Reviewer #1: No

3. Have the authors made all data underlying the findings in their manuscript fully available?

Reviewer #1: Yes

4. Is the manuscript presented in an intelligible fashion and written in standard English?

Reviewer #1: Yes

5. Review Comments to the Author

Reviewer #1: This article provides interesting data on faculty perceptions of publishing. This is a valuable and timely area of study. However, the paper would need to undergo significant revisions in order to meet the PLOS ONE publishing criteria (namely: (3) Experiments, statistics, and other analyses are performed to a high technical standard and are described in sufficient detail; and (4) Conclusions are presented in an appropriate fashion and are supported by the data.) I describe these concerns as well as other comments on improving the manuscript below.

The introduction assumes a North American context for academe, without explicitly framing it as such. It specifically references the growth of English-language journals and describes scenarios that apply predominately to North American universities. By taking for granted this context, it frames North American as the normative context for academic structure. I would strongly encourage the authors to reframe the introduction to acknowledge this context explicitly, rather than implicitly.

The last three research questions acknowledge that this is perception-based. The first question should also acknowledge that these are self-reported factors; the actual influence of these issues is difficult to measure post-hoc and via self-report. The authors should include a limitation section that acknowledges that what authors report as their rationale for choosing a publication may be an idealized version of reality. This may be another explanation for the “mismatch” observed.

What exactly is meant by “pseudo-randomly”? Why manual selection rather than the assignment of a number to each faculty member and the use of a random number generator? This description of the sampling procedure reduces confidence.

Furthermore, and perhaps more importantly, it is unclear why the sampling used the sample frame of institutions derived from the previous study (Alperin et al., 2019). Given that no analysis was done linking responses with policies, there is no justification for this sample, which may not have been the most appropriate frame for the research questions posed. More justification for this choice is necessary.

The authors need to use caution in the interpretation and reporting of Likert-type questions. Means should not be used for not ratio data. For example, the authors examine the number of publications per year, in binned categories. They then conclude that women publish fewer articles than men—citing that their mean is 3.05 compared to 3.28, where a score of 3 indicates 1-2 publications and 4 indicates 3-5 publications. The authors should only use modal amounts and categorical analyses such as Chi-squared for these data.

In Table 2, it is essential to note that two things are skewing this: the first is that one-third of the respondents are not from research-intensive universities, yet they may consider peers to be at all universities. Therefore, it may not be necessarily incorrect that things like funding and JIF have higher importance to their peers. (As a small note of concern, the authors round 32.4% up to 33% in the description of the respondents; this is non-standard and should be fixed.) Furthermore, the majority of respondents are past-tenure; therefore, it again might be true that these things matter less. This may be less of a gap between perceptions of self and perceptions of others, but rather the fact that the selves and the peers are not equally represented in these data. I do not find convincing the statement that there is a “mismatch between faculty’s own values and how they perceive those of their peers”. The second paragraph of the discussion seems rather to support the interpretation that this is a mismatch of institutional culture and academic status. The entire discussion, however, seems to focus on this disconnect, which does not appear to be substantiated. The logic of the discussion seems highly problematic in that it at once acknowledges a difference in expectations by career stage (which is well-documented) and then suggests that junior faculty are somehow living under a false reality because they have different expectations and values than their tenured peers.

The authors fail to cite the most relevant and contemporary research in the introduction and discussion discussion. For example, in the discussion of gendered differences in the responses, they fail to note the considerable research on funding/resources (e.g., Ginther, Duch), attrition, or the productivity puzzle. The majority of their references are more than a decade old. In fact, they only cite four journal articles other than their own published in the last decade. For another topic, this would be irrelevant; however, given that they are making claims about the state of contemporary research and that more recent articles are available, I would encourage them to conduct an appropriate review of the literature. This will allow them to better motivate the article, to contextualize it in previous knowledge, and to demonstrate the contribution of this work.

Related to this, the authors state that the work builds upon Alperin (2019), but this is not listed in the references. Given that the reader is referred to this for an explanation of the methods, this is a major concern. (The work cannot be evaluated without the information in this reference and it is not available in the manuscript.)

6. PLOS authors have the option to publish the peer review history of their article (what does this mean?). If published, this will include your full peer review and any attached files.

Reviewer #1: No

---

## [Author Response · Author response to Decision Letter 0]

25 Nov 2019

Dear Editor and Reviewers, 

Thank you and the three reviewers for the helpful feedback on our manuscript “Why we publish where we do: Faculty publishing values and their relationship to review, promotion and tenure expectations”. We are pleased to resubmit our manuscript with the requested revisions. Below you will find the reviewers’ feedback (in gray) interspersed with a description of how we addressed each of the points raised (in black). 

As requested over email, please update the competing interests to read as follows: "MTN is a member of the board of directors of The Public Library of Science (PLOS). This role has in no way influenced the outcome or development of this work or the peer-review process, nor does it alter our adherence to PLOS ONE policies on sharing data and materials."

We would like to take this opportunity to express our appreciation to the reviewers for you’re their thoughtful feedback. We are convinced that we have adequately addressed all of the expressed concerns and that the manuscript has been improved as a result of this process.

Sincerely, 

Juan Pablo Alperin

Assistant Professor, Publishing

Associate Director, Public Knowledge Project

Director, Scholarly Communications Lab

Simon Fraser University

---

## [Decision Letter · Decision Letter 1]

28 Jan 2020

Why we publish where we do: Faculty publishing values and their relationship to review, promotion and tenure expectations

PONE-D-19-21675R1

Dear Dr. Alperin,

We are pleased to inform you that your manuscript has been judged scientifically suitable for publication and will be formally accepted for publication once it complies with all outstanding technical requirements.

With kind regards,

Sergio A. Useche, Ph.D.

Academic Editor

PLOS ONE

Additional Editor Comments (optional):

Reviewers' comments:

Reviewer's Responses to Questions

**Comments to the Author**

1. If the authors have adequately addressed your comments raised in a previous round of review and you feel that this manuscript is now acceptable for publication, you may indicate that here to bypass the “Comments to the Author” section, enter your conflict of interest statement in the “Confidential to Editor” section, and submit your "Accept" recommendation.

Reviewer #2: All comments have been addressed

2. Is the manuscript technically sound, and do the data support the conclusions?

Reviewer #2: Yes

3. Has the statistical analysis been performed appropriately and rigorously? 

Reviewer #2: Yes

4. Have the authors made all data underlying the findings in their manuscript fully available?

Reviewer #2: Yes

5. Is the manuscript presented in an intelligible fashion and written in standard English?

Reviewer #2: Yes

6. Review Comments to the Author

Reviewer #2: Good concept; fairly well designed study; well written and presented...

I do not agree to the statement that females submit less publications because they are more into teaching....It is said that ' One who does no research has nothing to teach '. Also, in most academic institutions, it is now mandatory that teachers should have to have a number of publications to their credit to make themselves eligible for promotion to higher posts.

7. PLOS authors have the option to publish the peer review history of their article (what does this mean?). If published, this will include your full peer review and any attached files.

Reviewer #2: No

---

## [Editor Report · Acceptance letter]

10 Feb 2020

PONE-D-19-21675R1 

Why we publish where we do: Faculty publishing values and their relationship to review, promotion and tenure expectations 

Dear Dr. Alperin:

I am pleased to inform you that your manuscript has been deemed suitable for publication in PLOS ONE. Congratulations! Your manuscript is now with our production department. 

With kind regards,

on behalf of

Dr. Sergio A. Useche 

Academic Editor

PLOS ONE